# Comparing Stakeholders' Economic Values for the Institution of Payments for Ecosystem Services in Protected Areas

**Namhee Kim [1], Miju Kim [2], Sangkwon Lee [3] and Chi-Ok Oh [4],***

1    Department of Cultural Studies, Chonnam National University, Gwangju 61186, Republic of Korea;
     186388@jnu.ac.kr
2    Marine Policy Research Department, Korea Maritime Institute, Busan 49111, Republic of Korea;
     mijukim@kmi.re.kr
3    College of Arts, Sciences, Business and Education, Winston-Salem State University,
     Winston-Salem, NC 27110, USA; lees@wssu.edu
4    Graduate School of Culture, Chonnam National University, Gwangju 61186, Republic of Korea
*    Correspondence: chiokoh@chonnam.ac.kr; Tel.: +82-62-530-4075

**Abstract:** In order to maintain the provision of high-quality ecosystem services in wetlands, it is important to protect the ecosystems through the designation of protected areas. However, the process of designating protected areas can potentially give rise to social conflicts or problems by the acquisition of private lands. As an alternative, the institution of payments for ecosystem services (PES) can be a more viable solution. This study intends to propose reasonable contract standards for PES that consider the preferences of both beneficiaries and providers, which are necessary for the successful introduction of PES in wetland protection areas in Korea. In doing so, we employed choice experiments to estimate the willingness to pay (WTP) and willingness to accept (WTA) of different stakeholders. Our findings indicate that both beneficiaries and providers had a positive perception of PES contract terms. Moreover, the WTP and WTA values were comparable, suggesting that the unit price of PES could be determined within a reasonable range. These results can serve as a foundation for acquiring additional funds required for the introduction of PES in wetland protected areas.

**Keywords:** protected areas; wetlands; payments for ecosystem services; choice experiments

## 1. Introduction

Wildlife habitats have been continually destroyed or degraded due to economic development and urban population concentration, adversely affecting biodiversity [1–3]. Many countries, including Korea, have designated protected areas to restore damaged ecosystems and prevent further deterioration [4]. These areas protect habitats, species, ecological processes, and organism functions, as well as provide ecosystem services such as water, food, air quality, and recreation [5]. In other words, designated protected areas safeguard nature and biodiversity and contribute to ecosystem services for human well-being [6].

In Korea, about 17% of the lands and 2% of the seas are protected areas. As of 2023, there are 44 protected wetlands accounting for 4% of the total protected areas in Korea. The criteria used to designate protected wetlands are (1) areas with primitive nature and ample biodiversity, (2) areas offering habitats for rare or endangered plant and animal species, and (3) areas with unique scenery or geological value [7]. The Korean government has been implementing a wetlands protection and management plan and trying to expand the protected areas from 1634.6 km$^2$ to 1730 km$^2$ by 2027.

Despite the government's efforts to protect these areas, there exist various problems, such as conflicts between private and social interests [8,9] and lack of management budgets [10,11]. In particular, 56% of the protected areas were designated during the 2000s, but most of them contain private lands, called inholdings. In Korea, wetland protected areas contain about 50% of private lands on average. Recently, one wetland site was designated

as a protected area despite 99% of private lands [12]. Due to the designations with considerable inholdings, there is a risk of conflicts between landowners pursuing economic benefits and the government trying to conserve the environment [13].

A possible resolution to this problem is the government purchasing private lands in protected areas. However, it would be infeasible for a government with a limited budget to pursue this option [14]. Another measure is that the government directly compensates the landowners commensurate with the economic benefits of ecosystem services. This measure, called payments for ecosystem services (PES), is an incentive mechanism that compensates landowners for providing ecosystem services in protected areas. The concept of PES has been applied in various environmental protection and conservation contexts. It is a mechanism for converting the values provided by ecosystem services into actual economic incentives [15].

PES is based on the beneficiary-pays principle, in which the beneficiary of an ecosystem service pays a certain amount of money to the service provider [16]. PES participants are composed of beneficiaries, providers, mediators, and local residents [17,18]. A beneficiary directly benefits from the ecosystem services, a provider performs specific actions to provide these services, and a mediator designs and implements PES by connecting beneficiaries and providers. Mediators include governments, associations, and institutions acting on their behalf. Local residents are those who live in protected areas and benefit from ecosystem services but do not participate in the system as direct beneficiaries or providers.

Financial support for introducing PES needs to be determined equitably using a compensation range that will benefit both beneficiaries and providers based on the principle of supply and demand [17]. For example, the beneficiary may be willing to choose a scheme with a lower price for ecosystem services than other alternatives. In contrast, the provider will only participate in PES if the money compensated offsets the reduced products or income. Therefore, the payment range may lie between a level that compensates the provider's loss from PES (the minimum level) and a level that imposes a price on the beneficiary's benefits from the ecosystem services (the maximum level).

It is important to implement PES that account for various stakeholders' interests to avoid any potential conflicts. For instance, if PES is implemented in such a way that only the provider's necessary payment for ecosystem services is considered, without taking into account the value of the service for the beneficiary, it may give rise to conflicts among stakeholders. Some PES studies have only focused on estimating the beneficiary's willingness to pay (e.g., [19–21]). In contrast, other PES studies have only estimated the provider's willingness to accept payment (e.g., [22–24]). However, different stakeholders should be considered in research as PES is a voluntary and conditional agreement between providers and beneficiaries [25].

Estimating the economic value of ecosystem services offers informative guidance on what price to charge to service users [26]. In addition to eliciting economic value, ensuring the beneficiaries and providers reach a mutual agreement on the prevailing amount is critical. This study aims to present the prevailing amount accounting for the preferences of beneficiaries and providers, which is essential to implementing PES in protected areas. The prevailing amount, including the payment range, is critical for selecting payment mechanisms. The estimated payment range reflects the economic value of ecosystem services in protected areas. This study estimates the beneficiary's willingness to pay (WTP) and the provider's willingness to accept (WTA) to suggest the acceptable amount range based on the WTP and WTA. The economic value of the protected area that this study estimated can be useful information for designing practical measures that focus on effectively implementing PES. Additionally, by providing both WTP and WTA, this study's findings can lead to in-depth discussions on WTP/WTA disparity, mitigating or resolving conflicts between stakeholders, and the practical implications associated with PES.

This study estimates the beneficiaries' WTP and providers' WTA by employing the choice experiment method and comparing the two values. The findings are expected to provide meaningful information for the PES of protected areas. The study site is the

Dongcheon Estuary Wetland Protected Area in Suncheon city, Korea, which was designated as a protected area in 2015 and still has a quite high proportion of private lands (53.3%). The farmers who participate in eco-friendly farming in this region have been supported through a direct payment program funded by the government. As the government plans to introduce PES instead of the direct payment program in the region, we chose this site and the adjacent region as a study site.

## 2. Literature Review

### 2.1. The Economic Valuation of Protected Areas

A protected area is "a clearly defined geographical space, recognized, dedicated and managed, through legal, or other effective means, to achieve the long-term conservation of nature associated with ecosystem services and cultural values" [27]. Protected areas can contribute to biodiversity conservation and people's livelihoods. The role and importance of protected areas require all government levels to actively initiate public policies to effectively manage and conserve these areas.

Estimating the economic value of protected areas has received considerable attention. The economic valuation of ecosystem services can be used to develop a payment mechanism by determining how much a beneficiary would be willing to pay and a provider would be willing to accept. Beneficiaries and providers should agree on the valuation and payments based on the opportunity cost associated with forgone revenues from ecosystem services.

Non-market valuation methods are commonly employed to assess the economic value of public goods, including tangible use and intangible non-use values. Stated and revealed preference methods are the most widely used in non-market valuation. Stated preference methods rely on individuals' responses to the hypothetical scenarios of proposed ecosystem services. The two main types of stated preference methods are the contingent valuation method (CVM) and choice experiments (CE). CVM is a method whereby survey respondents are asked to indicate their willingness to pay for non-market goods and services, such as protected areas, wetlands, forests, and recreational resources. It estimates the non-market goods and services' use value, existence value, option value, or bequest value [28].

CE typically presents respondents with multiple choice sets containing hypothetical alternative ecosystem services and monetary costs and asks them to choose the most preferred set of alternatives. The alternatives consist of attributes comprising two or more levels [29]. Ecosystem service values are inferred based on respondents' choices [30,31]. Thus, CE provides information about the amount individuals would pay for an attribute level improvement or the required compensation if an attribute level declines [32]. CE has been employed extensively to estimate the economic value of wetlands [33], biodiversity conservation [34], forest conservation and protection [35], and recreational resource management [36,37].

### 2.2. Payment for Ecosystem Services

The payment for ecosystem services (PES) has been widely implemented to encourage the conservation of protected areas, including wetland management, watershed management, and forest regeneration [18,38]. PES refers to a contractual transaction between a user and a provider for an ecosystem service or the land use/management practice likely to secure that service [39]. As most PES contracts are government payment programs aimed at enhancing social benefits, Muradian et al. [40] suggested a PES definition that reflects the Pigouvian conceptualization, which focuses on the public goods characteristics of ecosystem services, and the resulting externalities that will be internalized when PES is implemented. In the Pigouvian-based approach, a third party, such as a government agency, acts on behalf of users (beneficiaries). Most government-led PES adopting the Pigouvian approach have been implemented by either imposing taxes on negative externalities or subsidizing positive externalities [41].

PES can be instruments for helping to maintain the multi-functional role of wetlands ranging from enhancing biodiversity to improving water quality to mitigating climate change by sequestering and storing carbon. A variety of PES contracts were also carried out in forest conservation [42], watershed management [43], and recreational resource management [44]. For instance, in Costa Rica, PES implementation makes landowners receive payments for carrying out contracted activities for forest conservation and planned timber production. After the introduction of the scheme, the country's forest cover has increased to about 50% from 21% in 1987. The European Union (EU) also supports farmers adhering to EU regulations on food safety, environmental protection, and animal welfare. The EU offers various payments under the Common Agricultural Policy (CAP) [45]. In the United States, landowners and agricultural producers can benefit through financial and technical assistance programs such as the Conservation Stewardship Program (CSP) and the Environmental Quality Incentives Program (EQIP) [46]. Other efforts attempted to improve the environment through state payments for environmentally friendly farming or ranching include Germany's Landscape Conservation Payment Scheme, France's Nestle Water Practice, and Mexico's PES. South Korea has tried to introduce PES in a few protected areas on a pilot basis to ensure conservation and sustainable utilization of protected areas. In Korea, the central government is primarily responsible for instituting the main principles of PES. However, in order to reflect local characteristics and provide ongoing education and outreach activities to local residents, an implementation council is typically composed of public officials, experts, and residents' representatives [47].

Most economic valuation studies on PES pay attention to estimating the users' WTP for improved environmental services (e.g., [43,44]). However, relatively few studies have examined service providers' WTA payments to modify their behaviors (e.g., [23,24,48]). For example, Li et al. [48] estimated the economic value of PES on a reservoir catchment using households' WTA. Naime et al. [23] assessed the economic value of ecosystem services associated with forest restoration. Pérez-Rubio et al. [24] estimated cattle farmers' WTA to identify the economic value of ecosystem services, including erosion control, water availability, and biodiversity. Chang and Shin [49] assessed forests' economic value by eliciting WTP (USD 500 per hectare) and WTA (USD 2000 per hectare). Oh, Jung, and Joo [50] determined the economic value of ecosystem services provided by temple-owned forests in Korean national parks using CVM. Each household's WTP was USD 4 per year, and the total economic value was estimated at USD 78 million per year. Most PES-related studies evaluated the WTP and WTA separately for protected areas, which could make it hard for stakeholders to settle payment issues for PES. This study elicited the beneficiary's WTP and the provider's WTA regarding protected areas to help stakeholders implement PES efficiently and effectively.

## 3. Methods

### 3.1. Study Site

The study site was the Dongcheon Estuary Wetland Protected Area in South Korea (Figure 1). The wetland was designated as a national wetland protected area in 2015 and is one of Korea's largest inland wetlands (with a total area of 539.3 ha). Since the site was recently designated as a wetland protected area, a significant portion (about 53.3%) of the site is privately owned yet. Farmers cultivate inside and around the protected area. Of the total area excluding wetland, rice paddies occupy a significant portion, accounting for 53%, and 0.1% for fields, 1.5% for forest. Some of these farmers have participated in a biodiversity management contract project since 2005 through eco-friendly farming by avoiding pesticide use for diverse species. In addition, they have participated in a non-harvesting preservation project, in which some rice is purposefully not harvested and left for birds. The government offers a direct compensation of KRW 13,460,000 (USD 11,538.7) [1] per hectare to farmers, who participate in the project. The area is a suitable study site as the government plans to introduce PES instead of the current direct payments in the area.

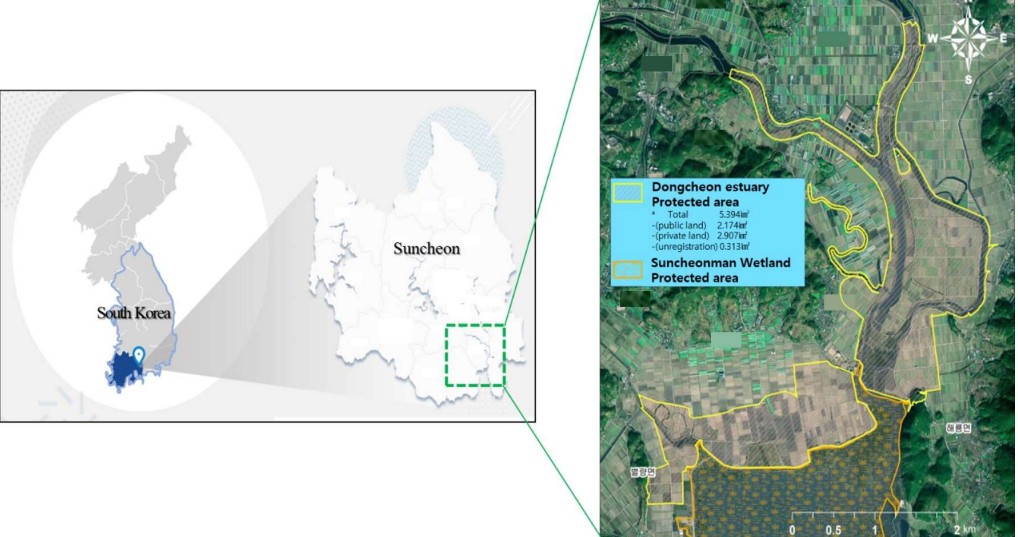

**Figure 1.** Study Site.

This wetland protected area holds 39 species of endangered animals, 848 species of wild animals, and wild birds including endangered ones such as spoonbills (*Platalea leucorodia*), storks (*Ciconia boyciana*), and hawks (*Falco peregrinus*) [51]. The Dongcheon stream flows into Suncheon Bay, leading to the Suncheon Bay Coastal Wetland Protected Area, a Ramsar Wetland and UNESCO Natural Heritage Site, which is considered a mudflat with great biological value. The Dongcheon Estuary Wetland and Suncheon Bay Wetland are adjacent; thus, they are regarded as the same area. Therefore, this study reasonably assumed that visitors to the Suncheon Bay Wetland and Dongcheon Estuary Wetland are the same. As Korea's representative wetland ecotourism site, annual visitors to this area amounted to about 1.95 million as of 2016.

*3.2. Choice Experiments*

To assess and compare WTP and WTA, we employed choice experiments (CE). Based on the random utility model, CE posits that respondent *i* maximizes their utility by choosing the better alternative *j* [52]. The indirect utility function *U* is composed of a deterministic component of utility *V*, which can be observed and is estimable, and a stochastic component of utility $\epsilon$, which is not directly observable to the researchers. *V* can be represented by an attribute vector, *X*, which is a vector of the attributes used in the choice set designs, and by the coefficient vector $\beta$ (i.e., parameter estimates).

$$U_{ij} = V_{ij}(X) + \epsilon_{ij} = X_{ij}\beta + \epsilon_{ij} \quad (j = 1, \ldots, j) \tag{1}$$

The probability that respondent *i* chooses alternative *j* is as follows in Equation (2):

$$P_{ij} = \Pr\left(V_{ij} - V_{im} \geq \epsilon_{im} - \epsilon_{ij} \forall j \neq m\right) \tag{2}$$

Assuming that all error terms follow the Type I extreme value distribution (or the Gumbel distribution) [53], Equation (2) can be presented as Equation (3) as a conditional logit model as follows:

$$P_{ij} = \frac{\exp(V_{ij})}{\sum_{j=1}^{J} \exp(V_{ij})} \tag{3}$$

A conditional logit model, as a basic CE model, does not consider the heterogeneity of respondents' preferences [54,55], only reflecting the preferences of the average respondent. Also, a conditional logit model's distributional assumption requires the satisfaction of independence from irrelevant alternatives (IIA) [56]. In contrast, the mixed logit model

(or random parameters logit model) estimates the coefficients of the attributes differently depending on the respondents. Also, as the mixed logit model relaxes the IIA assumption, multiple parameter estimates, which provide the random parameter distribution's mean and variance, explain the unobserved heterogeneity in the samples. Consequently, the mixed logit model was used in this study.

In the mixed logit model, the probability is expressed as Equation (4) [56].

$$P_{ij} = \int L_{ij}(\beta) f(\beta) d\beta, \ L_{ij}(\beta) = \frac{\exp(V_{ij}(\beta))}{\sum_{j=1}^{J} \exp(V_{ij}(\beta))} \tag{4}$$

$L_{ij}(\beta)$ is the estimated logit probability in parameter $\beta$. $f(\beta)$ is the probability density function, continuous in the mixed logit. If $f(\beta)$ is normalized by mean $b$ and covariance $W$, Equation (4) can be presented as Equation (5).

$$P_{ij} = \int \left( \frac{\exp(V_{ij}(\beta))}{\sum_{j=1}^{J} \exp(V_{ij}(\beta))} \right) \Phi(\beta|b, W) d\beta \tag{5}$$

When parameter $\beta$ is normally distributed with mean $b$ and covariance $W$, the researcher's goal is estimating $b$ and $W$. The economic value of policy changes or ecosystem services with choice experiments can be estimated as the marginal value of the changes caused by each attribute. Marginal willingness to pay (MWTP) is estimated by dividing the parameter estimates of the $k$th attributes $\hat{\beta}_k$ by the negative parameter estimates of monetary attribute $\hat{\beta}_u$. The marginal willingness to accept (*MWTA*) is estimated by dividing the parameter estimates of the $k$th attributes $\hat{\beta}_k$ by the parameter estimates of monetary attribute $\hat{\beta}_u$.

$$MWTP = -\frac{\hat{\beta}_k}{\hat{\beta}_u}, \ MWTA = \frac{\hat{\beta}_k}{\hat{\beta}_u} \tag{6}$$

### 3.3. Study Designs

The private lands of the Dongcheon Estuary Wetland Protected Area have been primarily used for agriculture. Thus, CE was designed to focus on promoting farming activities for ecosystem services. We performed an extensive literature review to select the important attributes of experimental designs (Figure 2). We reviewed programs such as the PES and similar systems that employed the choice experiment designs. Through this process, 15 attributes were initially considered but were further reduced to five (Table 1) based on discussions with researchers and managers focused on the feasibility of PES. Then, these attributes were further reviewed by the field managers of the protected area. Finally, a pilot test was conducted with 216 members of the general public to revise the attributes and levels.

Table 1 shows the five attributes used in the choice experiments. The first four attributes indicate the contracts that the providers should comply with, and the other attribute is the amount to be paid by the beneficiary. These contract-related attributes include "water quality improvement (QUAL)", "removal of invasive alien species (ALIEN)", "biodiversity conservation activity (DIV)", and "technical advice (ADV)" (Table 1).

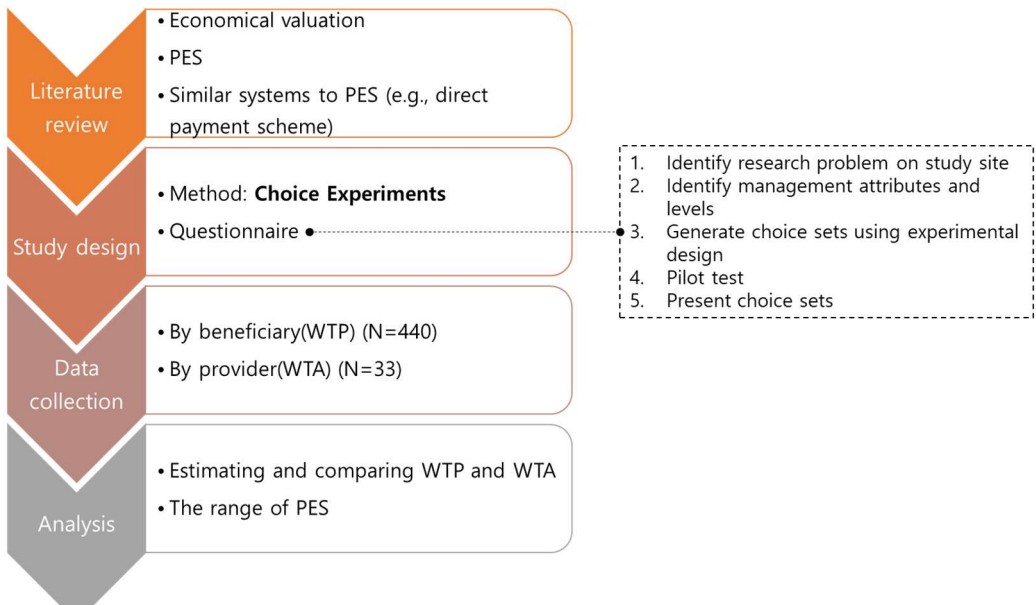

**Figure 2.** Process of the study.

**Table 1.** Attributes and levels used in CE.

| Attributes | Descriptions | Levels |
|---|---|---|
| QUAL | The use of fertilizer to improve water quality by reducing fertilizer to decrease nitrogen (pollutant) emissions | -use of 33% fertilizer<br>-use of 66% fertilizer<br>-use of 100% fertilizer * |
| ALIEN | Farmers' removal of alien species to help reduce their negative impacts on ecosystems. | -participation<br>-non-participation * |
| DIV | Farmers' installation of natural drainage or trap water in irrigation channels and drain systems in paddy fields in winter for biodiversity conservation. | -participation<br>-non-participation * |
| ADV | Educating farmers on PES and implementing farming practices | -with education<br>-lacking education * |
| (for the beneficiary) Additional income tax | Additional income tax per household per year for benefits from provider's PES contract activities | -(0 *),<br>-KRW 1000(USD 0.9)<br>-KRW 3000(USD 2.6)<br>-KRW 5000(USD 4.3)<br>-KRW 7000(USD 6.0)<br>-KRW 10,000(USD 8.6) |
| (for the provider) Subsidy | Annual subsidy received on the condition that providers carry out PES contracts | -(0 *),<br>-KRW 1 million (USD 858.1),<br>-KRW 2 million (USD 1716.2),<br>-KRW 3 million (USD 2574.3),<br>-KRW 5 million (USD 4290.5),<br>-KRW 7 million (USD 6006.5) |

* Current status.

The first attribute is QUAL, which refers to using less fertilizer, and what is used is eco-friendly rather than chemical. When fertilizer is reduced, fewer pollutants, such as nitrogen, will flow into the river, improving water quality. This attribute used the following three levels: use of 33% fertilizer (emissions of 50% of the current pollutant level), use of 66% fertilizer (emissions of 75% of pollutants), and use of 100% fertilizer (emissions of 100% of pollutants). The second attribute is ALIEN, which means the removal activities of alien species that are a threat to native species to reduce their negative impacts on the ecosystem. The two levels used to determine this are "participation" and "non-participation". The third attribute is DIV, which refers to activities that enhance biodiversity by conserving the habitats of organisms around the wetland. These activities include installing natural drainage or trapping water in irrigation channels and drain systems in paddy fields in the wintertime. This is also determined using the two levels of "participation" and "non-participation". The fourth attribute is ADV, which is the provision of information to farmers so they can understand PES and practice appropriate farming methods, which is expected to promote farmers' participation in PES. The two levels used to determine this are "with education" and "lacking education". The final attribute is a payment vehicle, set differently depending on beneficiaries and providers. The payment vehicle for beneficiaries was an additional income tax per household per year. Five levels were employed through a literature review, and preliminary tests determined the payment levels from KRW 1000 (USD 0.9) to KRW 10,000 (USD 8.6). The payment vehicle for providers was an annual subsidy per unit area. The levels included subsidies of KRW 1 million (USD 858.1), KRW 2 million (USD 1716.2), KRW 3 million (USD 2574.3), KRW 5 million (USD 4290.5), and KRW 7 million (USD 6006.5) per hectare. The base levels of the attributes, set to the current level in Table 1, were the use of 100% fertilizer, non-participation in the removal of invasive species, non-participation in biodiversity conservation activities, lacking technical advice, and zero additional income tax or subsidy.

An efficiency design [57] was employed to generate a manageable number of paired choice sets (i.e., 30 paired choices for beneficiaries and 36 paired choices for providers) using five attributes in Table 1 and their levels. The 30 choice sets were further divided into five blocks (i.e., different versions of the questionnaire) for beneficiaries, and the 36 choice sets were divided into four blocks for providers. Therefore, six paired choice sets, consisting of three options (i.e., two alternatives and no-choice), were presented to each respondent of the beneficiary group, and nine paired choice sets were presented to the provider group. An example of the paired choice sets is shown below in Figure 3 (see more detailed explanations on the attributes and levels in the Supplementary Materials).

*3.4. Data Collection*

This study's beneficiaries included visitors to the Dongcheon Estuary Wetland Protected Area and nearby residents over the age of 20. On-site and online surveys were conducted with Suncheon Bay Natural Ecology Center visitors from August to December 2019. Those who preferred an online survey were emailed a link to SurveyMonkey®®. For residents who lived in the Dongcheon Estuary Wetland Protected Area and the adjacent counties, an online survey was conducted using a panel recruited by a survey company (Embrain, Seoul, Republic of Korea).

This study's providers were those who lived around the Dongcheon Estuary Wetland Protected Area and were supported by direct payment for eco-friendly farming. An on-site survey was conducted with 52 providers from October to November 2019. The survey respondents comprised 474 beneficiaries and 52 providers. However, data provided by 34 beneficiaries and 19 providers were discarded due to missing values and unqualified responses. Finally, data from 440 beneficiaries and 33 providers were used for the analysis.

※ The Korean Government plans to implement the payment of ecosystem services (PES) to manage private lands (rice paddies) in an eco-friendly manner within the Dongcheon Estuary Wetland Protected Area in Suncheon. Implementing this policy will benefit residents and visitors as the Dongcheon Estuary Wetland will be better managed. However, to support this, they will need to pay taxes to the extent of the benefits to the government. From this, the government provides subsidies to the growers (providers) who participate in eco-friendly farming practices. Through this policy, the providers can be compensated for the potential financial losses caused by participation in environmentally friendly farming practices, and the beneficiaries can continually enjoy the benefits from the conservation of the environment, such as clean water and well-protected scenery.

*(Survey for beneficiaries)* Given only the three alternatives below for providers to participate in PES, which would you prefer? (Assuming that a change in one attribute does not affect the others).

*(Survey for providers)* Which of the following alternatives would you prefer for participating in PES?

| Attributes | Alternative 1 | Alternative 2 | Alternative 3 |
|---|---|---|---|
| **Water quality improvement** | use of 100% fertilizer (emissions of 100% of pollutants) | use of 33% fertilizer (emissions of 50% of pollutants) | |
| **Removal of alien species** | 외래종 non-participation | 외래종 participation | |
| **Biodiversity conservation activities** | participation | non-participation | I would not choose either option |
| **Technical advice** | lacking education | with education | |
| (For beneficiaries) **Additional income tax** (For providers) **Subsidy** | KRW 3000 KRW 3 million | KRW 1000 KRW 1 million | |
| **Choice** | ☐ | ☐ | ☐ |

**Figure 3.** An example of paired choice sets.

## 4. Results

### 4.1. Descriptive Statistics of the Respondents' Sociodemographics

The respondents' descriptive statistics are shown in Table 2. Of the beneficiaries and providers, the proportion of women was 59% and 31%, and the mean age was 42 and 67, respectively. Approximately 73% of beneficiaries had completed a college degree or higher, whereas 3% of providers were in the same category. This indicates that the providers were comparatively older and less educated than the beneficiaries. Of the beneficiaries, 45% were aware that the study site was a wetland protected area. The average number of visits to the site over the past five years was 3.3, and about 61% of the beneficiaries were from the same province where the study site is located.

**Table 2.** Descriptive statistics.

| Descriptions | Beneficiary | Provider |
|---|---|---|
| Total respondents | 440 | 33 |
| Female (%) | 59.2 | 31.4 |
| Age * | 41.9 (13.8) | 67.0 (7.8) |
| Monthly household income per family member * | USD 1574.3 (956.0) | USD 441.3 (427.4) |
| Education (% of college degree or higher) | 72.5 | 3.0 |
| Awareness of protected area (%) | 44.9 | . |
| Number of visits during the last five years * | 3.3 (9.3) | . |
| Regional origin | | |
| The province in which the study site is located | 268 (60.9%) | |
| Otherwise | 164 (37.3%) | |
| No response | 8 (1.8%) | |

* Parentheses indicate the standard deviation.

We explained the ecosystem services provided in the site to the respondents. Regarding which group would benefit most from the ecosystem services, 42% of the beneficiaries answered "all citizens", 38% answered "local residents", and 16% answered "visitors". In the questionnaire for providers, the majority of providers (57%) reported that the benefits provided by the study site were similar to or slightly better than before. Additionally, half of the providers (50%) answered that the local residents are the primary group responsible for changes in the provision of ecosystem services.

*4.2. Analysis of the Mixed Logit Model*

We analyzed the data using a mixed logit model developed by Hole [58] for both beneficiaries and providers (Table 3). We used Stata16.0 and 500 Halton draws for data analysis. An alternative specific constant (ASC) was inserted to capture the attribute effects not included in the utility function. Dummy coding was used for qualitative attributes except those of Tax/Subsidy. For example, the attribute of QUAL with the three levels was estimated with two dummy-coded variables (i.e., QUAL1 for using 33% fertilizer and QUAL2 for using 66% fertilizer). As using 100% fertilizer was the base option, the positive coefficients of QUAL1 and QUAL2 indicated the respondents' favorable preferences for those two options compared to the base option. We also calculated McFadden's $\rho2$ to assess the model's explanatory power and checked for heterogeneity in respondent preferences using the standard deviation of the coefficients.

Table 3 shows the estimation results. A goodness-of-fit measure, McFadden's $\rho2$, indicated 0.08 for the beneficiary group and 0.06 for the provider group. All variables were statistically significant, at least at the 1% level, except for the ASC of the beneficiary group. For the provider group, all variables were statistically significant, at least at the 10% level, except for DIV. The coefficients of the variables had the expected signs. The positive coefficients of QUAL1, QUAL2, ALIEN, DIV, and ADV indicate that the respondents preferred to improve water quality, remove alien species, conduct biodiversity activities, and receive technical advice. A negative coefficient of "income tax" in the beneficiary group denotes that they did not want a tax increase. However, a positive coefficient of "subsidy" in the provider group means that the respondents want to increase subsidies. Also, the significant standard deviations of the coefficients suggest that incorporating individual heterogeneity into the model is beneficial.

*4.3. WTP and WTA Estimation*

To estimate the respondents' preferences for each attribute as an economic value, each attribute's coefficient was divided by the payment vehicle coefficient to obtain the *MWTP* and *MWTA* (Table 4). The bootstrap method was used to calculate 95% confidence intervals. As the payment vehicle was answered per household, we converted it to the value per person using 2.5, the average number of household members [59].

**Table 3.** Estimation results.

| Attribute | Beneficiary | | | Provider | | |
|---|---|---|---|---|---|---|
| | Coefficient | Std. Err | Std. Dev. of Coefficient | Coefficient | Std. Err | Std. Dev. of Coefficient |
| ASC [1] | 0.1190 | 0.126 | | 1.2605 ** | 0.515 | |
| tax/subsidy [2] | −0.1397 *** | 0.014 | | 0.2934 *** | 0.061 | |
| QUAL1 [3] | 1.6505 *** | 0.132 | 1.5246 *** | 0.8144 ** | 0.393 | 1.3762 *** |
| QUAL2 [4] | 0.9313 *** | 0.111 | 0.9577 *** | 0.4888 * | 0.255 | 0.1576 |
| ALIEN [5] | 0.3220 *** | 0.100 | 1.2554 *** | 0.6107 * | 0.333 | 1.3902 *** |
| DIV [6] | 0.6781 *** | 0.098 | 1.1622 *** | 0.3106 N.S. | 0.237 | 0.6396 * |
| ADV [7] | 0.6429 *** | 0.105 | 1.4011 *** | 0.5182 ** | 0.240 | 0.203 |
| | | N: 2631 | | | N: 297 | |
| Model fit | Log Likelihood: −2356.5408 Pseudo-R$^2$: 0.0752 | | | Log Likelihood: −204.6419 Pseudo-R$^2$: 0.0553 | | |

Significance levels of 0.1, 0.05, and 0.01 are represented by *, **, and ***, respectively. N.S. means "not significant". [1] ASC means alternative specific constant. [2] Tax is per thousand KRW, and subsidy is per million KRW. [3] QUAL1: water quality improvement 1. [4] QUAL2: water quality improvement 2. [5] ALIEN: removal of alien species. [6] DIV: biodiversity conservation activities. [7] ADV: technical advice.

**Table 4.** Results of the WTP and WTA estimation.

| Attributes | Beneficiary | | | Provider | | |
|---|---|---|---|---|---|---|
| | *MWTP* [1] | 95% CI [3] | | *MWTA* [2] | 95% CI | |
| QUAL1 | 4.1 | 2.9 | 5.2 | 2382.0 | −880.4 | 5643.8 |
| QUAL2 | 2.3 | 1.5 | 3.1 | 1429.6 | −296.9 | 3156.1 |
| ALIEN | 0.8 | 0.3 | 1.3 | 1785.7 | −755.1 | 4327.4 |
| DIV | 1.7 | 1.1 | 2.2 | 908.7 (NS) [4] | −1094.1 | 2910.7 |
| ADV | 1.6 | 1.0 | 2.2 | 1515.4 | −191.4 | 3222.2 |

[1] *MWTP* is in USD per person per year. [2] *MWTA* is in USD per hectare per year. [3] Confidence intervals (CI) were obtained with 500 bootstrap replications. [4] NS means "not significant".

Assuming the other attributes remain the same (*Ceteris paribus*), the annual *MWTP* of beneficiaries per person was about KRW 4726 (USD 4.1) for QUAL1, KRW 2667 for QUAL2 (USD 2.3), KRW 922 (USD 0.8) for ALIEN, KRW 1942 (USD 1.7) for DIV, and KRW 1841 (USD 1.6) for ADV. These values indicate the maximum additional amount beneficiaries are willing to pay per person if the providers fulfill the above PES contracts. For example, the *MWTP* of QUAL1 suggests that beneficiaries are willing to pay USD 4.1 for the 33% use of fertilizers instead of 100%.

The provider's *MWTA* per hectare was about KRW 2.78 million (USD 2382.0) for QUAL1, KRW 1.67 million (USD 1429.6) for QUAL2, KRW 2.08 million (USD 1785.7) for ALIEN, KRW 1.06 million (USD 908.7) for DIV, and KRW 1.77 million (USD 1515.4) for ADV. These values indicate the minimum subsidy per hectare of private land providers are willing to accept for executing PES contracts as above. For example, the *MWTA* of QUAL1 shows that providers who used 100% fertilizer are willing to reduce it to 33% if they receive a subsidy of at least USD 2382.0 per hectare.

The results of the *MWTP* and *MWTA* show that beneficiary and provider groups preferred QUAL1 (Table 4). This suggests that both groups consider QUAL1 the most valuable PES contract activity. Beneficiaries preferred QUAL2 after QUAL1, but providers considered ALIEN and ADV more important than QUAL1. DIV was important for beneficiaries, but the derived attribute value was not significant for providers.

*4.4. Assessing Management Scenarios*

Based on the changes in water quality improvement levels, we developed two scenarios: "QUAL 1" and "QUAL 2". The "QUAL 1" scenario involves reducing fertilizer use from 100% to 33% and participating in the removal of alien species, biodiversity conservation, and technical advice and training. The "QUAL 2" scenario also involves reducing fertilizer use, but only to 66%, and participating in the same initiatives for the removal of

alien species, biodiversity conservation, and technical advice and training. The study site is located within a designated wetland protection area and is renowned for its eco-friendly farming practices that cater to migratory bird populations. Given the high expectations of both the beneficiaries and providers towards eco-friendly farming methods, the ecosystem service payment was calculated under the QUAL1 scenario, involving a significant reduction in fertilizer usage.

In the QUAL1 scenario, all attributes were significant in the beneficiary's data, so the willingness to pay was calculated, including all attributes. However, since DIV was not significant in the provider's data, the willingness to accept payment was calculated, excluding this attribute. The beneficiary was willing to pay KRW 23,578 (USD 20.2) per household per year. Likewise, the provider was willing to accept a subsidy of KRW 6.62 million (USD 5683.0) per hectare per year.

WTP and WTA are not directly comparable due to the difference in their units. However, they could be compared after making the units of the two values the same through the following process. First, the beneficiary group includes visitors and local residents. The number of households visiting the Ecology Center was 216,155 (total number of visitors ÷ average number of households). Since people can visit both the Suncheon Bay Coastal Wetland Protected Area and Dongcheon Estuary Wetland Protected Area at this eco-center, the number of households visiting the Dongcheon Estuary Wetland Protected Area was 126,115, calculated by dividing the number of visitors by the ratio of the area of the two protected areas (6.2). Next, the total number of households that reside adjacent to the protected area was 21,205 as of 2015. Therefore, the total number of beneficiary households, including both local residents and visitors, was 147,320.

Then, we multiplied the number of households by the beneficiary's willingness to pay KRW 23,578 (USD 20.2) per household per year, resulting in an annual beneficiary benefit of about KRW 3.47 billion (USD 2.98 million). Dividing this value by the area of the Dongcheon Estuary Wetland Protected Area (539.4 hectares) produced an annual WTP of KRW 6.44 million (USD 5526.9) per hectare. The beneficiary's WTP value was similar to the provider's WTA of KRW 6.62 million (USD 5683.0) per hectare per year.

## 5. Discussion of the Results

### 5.1. PES Contract Preference

This study aimed to compare beneficiaries' *MWTP* and providers' *MWTA* to suggest a reasonable compensation amount for PES. To achieve this purpose, a compensation amount was derived using CE that reflects the beneficiaries' and the providers' preferences. The study results show that both beneficiary and provider groups preferred water quality improvement. The results correspond with the findings of previous studies on the wetland ecosystem service preferences of PES contracts [60–63]. As water quality is a major contributor to various ecosystem services, from outdoor recreation activities to health [64], it is rational that both beneficiaries and providers perceived this attribute as the most important one. Regarding the other attributes, the preferences of the beneficiaries and providers were quite different. While the beneficiaries preferred biodiversity conservation to removing alien species, the providers evaluated removing alien species as more important than biodiversity conservation. The providers were likely to value removing alien species more because their participation in this activity was directly related to their income. In contrast, most beneficiaries, as visitors to the protected areas, were more interested in the benefits of cultural ecosystem services, such as recreation, landscape, and education, than those derived from provisioning and regulating ecosystem services, such as food or climate control [65]. Therefore, the beneficiaries placed more importance on biodiversity conservation activities than removing alien species.

### 5.2. WTP and WTA Comparison for PES

The study site is a designated protected area. In some parts of the site, eco-friendly farming methods have already been implemented. Therefore, WTP and WTA were derived

based on the "Water Quality Improvement 1" scenario, which was considered feasible. The results of the scenario analysis show that the beneficiary's WTP was KRW 23,578 per household per year (USD 20.2). The beneficiary's WTP was converted to the same monetary unit per hectare for comparison with the providers' WTA, resulting in KRW 6.44 million (USD 5526.9) per hectare per year. The provider's WTA was KRW 6.62 million (USD 5683.0) per hectare per year. For comparisons, Switzerland's ecological payments provided up to USD 2225/ha for pesticide-free, organic, and other eco-friendly farming practices. In Bavaria, Germany, landscape conservation payments provided up to USD 2457/ha for landscape conservation [66]. Costa Rica's PES compensated participants USD 210–327/ha, depending on the contract [67]. A study by Johnson et al. [68] indicated that benefits provided by Conservation Reserve Program (CRP) lands were worth USD 692–2591/ha, exceeding the cost of payments to farmers. This study shows a range of USD 547.5–2800, depending on the attribute, when the *MWTP* was converted to the unit per hectare per year. Therefore, the range of PES amounts estimated in this study are generally in a similar range to the amount derived from similar PES cases. Consequently, the beneficiary's WTP and the provider's WTA were within a comparable range, indicating that both groups agreed to introduce PES as a conservation policy tool.

In general, a compensation amount for PES needs to be determined between the provider's WTA and the beneficiary's WTP. However, the study shows the provider's WTA slightly exceeded the beneficiary's WTP, which may contradict the premise of PES. In previous studies that compared WTP and WTA, Pagdee and Kawasaki [69] and Barr and Mourato [70] obtained similar results to this study, showing that WTA was greater than WTP. The phenomenon presumably resulted from the difference in the meaning of cost perceived by the beneficiaries and the providers. For the beneficiaries, the cost is to improve the benefits of cultural services when visiting the area, but for the providers, it can be an additional cost or work with extra labor [70]. Thus, financing or compensation from the government is required to resolve the imbalance between supply and demand [69,70].

## 6. Conclusions

This study estimated the economic values of ecosystem services provided by protected areas based on the beneficiary-pays principle for introducing PES. We employed CE to assess the *MWTP* of the beneficiaries and *MWTA* of the providers. The results imply that both groups positively perceived the introduction of PES.

Based on the results, the following implications are worth noting. First, the beneficiary's and provider's substantial economic values of PES indicate that they are aware of the need for PES. In particular, the beneficiary's WTP can be considered a reflection of the social benefit of PES. Therefore, WTP can be used by the government to implement PES by levying taxes on beneficiaries and providing subsidies to providers [71].

Second, the beneficiaries' WTP and the providers' WTA were within a comparable range. Subsequently, given that the providers' compensation amount is key to their decision to participate in PES, the results likely support that the unit price of PES can be determined within an acceptable range. While previous studies have focused solely on either the beneficiaries' WTP or the suppliers' WTA, this study intended to overcome this limitation by estimating and comparing both the providers' WTA and the beneficiaries' WTP. The results indicate that beneficiaries' WTP can be used as a financial resource for introducing PES and as a basis for unit price negotiations with the provider's WTA as the maximum range. However, if the providers' subsidy needs to be the WTA value or higher, additional resources equal to the difference between WTA and WTP will be required. Therefore, the results can be used as a basis for designing government funds to provide additional resources. Currently, in the Dongcheon Estuary Wetland Protected Area, several government subsidy policies are being implemented through direct compensation schemes to the providers, who participate in eco-friendly farming methods and non-harvest preservation projects. These scattered policies may hinder their efficiency. Thus, if PES is introduced to integrate

these policies, it will be possible to implement the contract more efficiently by suggesting a reasonable cost to both the beneficiary and the provider.

Several limitations can help guide future research. First, this study focused on the benefits and provisions of ecosystem services through the fulfillment of the PES contract. Thus, the changes in ecosystem service levels of the PES contract were not clearly presented. Due to a lack of existing research on assessing ecosystem services in protected areas, this has not been supplemented so far. With additional studies conducted in the future, data will be accumulated. This will help us understand the beneficiaries' WTP and the providers' WTA more accurately. Second, due to the distinctive characteristics of the providers (e.g., age, education), a relatively incomplete number of them were aware of the purpose of PES and participated in the data collection. Providing ongoing public outreach and education programs will be beneficial to help increase their understanding and participation. Also, researchers need to consider different survey methods that can help them participate in data collection. Third, this study investigated the feasibility of introducing PES based on the case of Dongcheon Estuary Wetland Protected Area. Thus, the findings are likely to be insufficient to generalize to PES, and future studies that focus on other study sites will be beneficial to generalize study results.

This study confirmed the providers' and beneficiaries' preferences for PES and identified a range of reasonable compensation unit prices for PES implementation. In protected areas, direct subsidies for farmers have already been implemented. Therefore, the beneficiaries of ecosystem services may not have been directly aware of their role in ecosystem protection. If PES is introduced, the beneficiaries' interest and perceived value of ecosystem services will increase. Through this process, it is also expected that the meaning of these protected areas designated for ecosystem conservation will be strengthened. Moreover, given that PES has been put forth as a means to remedy conflicts between different stakeholders within protected areas [9,13,72], study findings can be also useful to other countries with similar challenges.

**Supplementary Materials:** The following supporting information can be downloaded at: https://www.mdpi.com/article/10.3390/land12071332/s1, Questionnaires.

**Author Contributions:** Conceptualization, C.-O.O., M.K. and N.K.; Methodology, C.-O.O. and M.K.; Formal analysis, M.K.; Writing–original draft, N.K., M.K., S.L. and C.-O.O.; Writing–review & editing, N.K., C.-O.O. and S.L. All authors have read and agreed to the published version of the manuscript.

**Funding:** We wrote this paper based on a research project titled "Economic valuation on Wetland Protected Areas for the Introduction of Payments for Ecosystem Services" funded by the Korea National Institute of Ecology. We appreciate Dr. Wooyeong Joo for his support.

**Data Availability Statement:** Data available on request due to privacy restrictions. The data presented in this study are available on request from the corresponding author.

**Conflicts of Interest:** The authors declare no conflict of interest.

## Notes

[1]   USD 1 = KRW 1165.4 (as of 2019, OECD).

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
