# Peer review of "Comparing Stakeholders’ Economic Values for the Institution of Payments for Ecosystem Services in Protected Areas"

_land, doi:10.3390/land12071332_

Round 1

Reviewer 1 Report

Dear Authors,

the paper presents an engaging research which brings some interesting insights to the current state of knowledge and fits to the scope of the journal Land.

The paper analyses beneficiaries’ MWTP and providers’ MWTA to suggest a reasonable compensation amount for PES and can help to avoid any potential conflicts. It is an important and current topic also in European Union. However, I have some comments which should be taken into consideration before publishing:

1.     The study area should be described more in detail. Please add information about: administrative division, land use (“most of the private land is used as rice paddies” – but the reader should get the data about the whole land use structure in %), population and especially number of farmer (also number of farmers who participated in a in a biodiversity management contract project since 2005), farm size etc. A more accurate landuse map would also be useful.

2.     What exactly means “Metropolitan area” and “the area near the study site”?

3.     Monthly household income - if such data have been collected during survey, it would be useful to provide the monthly household income per household member. Statistical sources, however, are much less accurate when we analyze the results of surveys (line 356-357). And it is the income per family member that often determines the willingness to pay. It is impossible to compare large families and the so-called singles.

4.     Please insert information, who ultimately introduces fees (PES) in Korea. Are these fees (rates) set at the government or local level? As some studies show, local authorities in democratic countries are often much less willing to introduce additional fees due to the fear of losing popularity. Fees managed centrally seem to be more effective (https://doi.org/10.3390/w14162453).

5.     It is worth recalling the exemplary amounts of ecosystem benefits in Korea in protected areas, but also refer to other examples, e.g. from the EU (EU Common Agricultural Policy (CAP) - https://doi.org/10.3390/su132313090, .

6.     Figure 2. Alternative 1 – for biodiversity conservation activities should be “non-participitation”

7.     Please correct the fonts in the description of the symbols (subsection 3.2).

8.     Maybe it's worth entering conclusions in bullets points.

Best regards.

Minor editing of English language required.

Author Response

The paper presents an engaging research which brings some interesting insights to the current state of knowledge and fits to the scope of the journal Land.

The paper analyses beneficiaries’ MWTP and providers’ MWTA to suggest a reasonable compensation amount for PES and can help to avoid any potential conflicts. It is an important and current topic also in European Union. However, I have some comments which should be taken into consideration before publishing:

  1. The study area should be described more in detail. Please add information about: administrative division, land use (“most of the private land is used as rice paddies” – but the reader should get the data about the whole land use structure in %), population and especially number of farmer (also number of farmers who participated in a in a biodiversity management contract project since 2005), farm size etc. A more accurate landuse map would also be useful.

-As suggested, we inserted additional information related to the study site in the manuscript.

  1. What exactly means “Metropolitan area” and “the area near the study site”?

-We originally meant that most respondents of the beneficiary group came from the same region (i.e., province) in which the study site is located. In order not to confuse readers, we deleted the phrase of metropolitan regions and indicated that over two-thirds of the beneficiaries were from the same province, in which the study site is located. Thank you.

  1. Monthly household income - if such data have been collected during survey, it would be useful to provide the monthly household income per household member. Statistical sources, however, are much less accurate when we analyze the results of surveys (line 356-357). And it is the income per family member that often determines the willingness to pay. It is impossible to compare large families and the so-called singles..

-We agree that income per family member would be important to determine the WTP. Thus, we changed the total household income to the income per family member (total household income / # of family members).

  1. Please insert information, who ultimately introduces fees (PES) in Korea. Are these fees (rates) set at the government or local level? As some studies show, local authorities in democratic countries are often much less willing to introduce additional fees due to the fear of losing popularity. Fees managed centrally seem to be more effective (https://doi.org/10.3390/w14162453).

-In Korea, the central government (mainly the Ministry of the Environment and Ministry of Agriculture) determines the rates but depending on programs and budgets, local governments also partially support it. We added this information in the payment for ecosystem services part (subsection 2.1.).

  1. It is worth recalling the exemplary amounts of ecosystem benefits in Korea in protected areas, but also refer to other examples, e.g. from the EU (EU Common Agricultural Policy (CAP) - https://doi.org/10.3390/su132313090, .

-As suggested, we added some examples from different countries (i.e., Costa Rica, EU, US) in subsection 2.1. We also indicated similar programs or schemes in other countries and compared study results to those in the WTP and WTA Comparison for the PES part (subsection 5.2.).

  1. Figure 2. Alternative 1 – for biodiversity conservation activities should be “non-participitation”

-Figure 2 is an example of the choice sets presented to the respondents. The levels used in the paired choice sets should be all different. However, in order to avoid confusion, we corrected the Figure

  1. Please correct the fonts in the description of the symbols (subsection 3.2).

-We corrected the fonts in the subsection 3.2.

  1. Maybe it's worth entering conclusions in bullets points.

-We have thought about this option but bullet points seem to be somewhat awkward. We hope it is OK not to make this change as suggested. Thank you.

Reviewer 2 Report

This study estimates the benefits’ WTP and providers’ WTA by employing the choice experiment method and comparing the two values. The content and structure of the article are reasonable, which is helpful to explore how to protect wetlands. However, the study  only relies on questionnaires to obtain data, and only 473 questionnaires were obtained, which greatly reduced the analyzability and reliability of the data. This is the primary reason for my doubts.

In addition, regarding the representativeness of the research area, that is, whether the conclusions of the research on the wetland can be used for reference by other countries. I think the international perspective of the manuscript is seriously insufficient, not only reflected in the introduction, so strengthening the comparison with wetlands in other countries can enhance the research significance of this article.

Finally, many details of the manuscript need to be supplemented, such as the chart design is not standardized (three-line table), the questionnaire design needs to be displayed, and the validity of the questionnaire, etc. Moreover, the language expression of the article needs to be improved, and further polishing is recommended.

Extensive editing of English language required

Author Response

Reviewer 2

This study estimates the benefits’ WTP and providers’ WTA by employing the choice experiment method and comparing the two values. The content and structure of the article are reasonable, which is helpful to explore how to protect wetlands. However, the study only relies on questionnaires to obtain data, and only 473 questionnaires were obtained, which greatly reduced the analyzability and reliability of the data. This is the primary reason for my doubts.

-Most CE studies have been conducted with 300-500 respondents (e.g., Ryffel, Rid, Grêt-Regamey, 2014; Hanley, Wright, Adamowicz, 1998; Dias, Belcher, 2015). According to Orme (2010), 200-300 respondents are typically recommended for choice experiments. While we collected the data with 473 respondents, this number should be in a reasonable range. In addition, because each respondent was asked to answer 6 paired choice sets, the actual observations were 2,631 for the beneficiaries and 297 for the providers after deleting some missing responses. The number of observations should be sufficient for data analysis. Thank you.

In addition, regarding the representativeness of the research area, that is, whether the conclusions of the research on the wetland can be used for reference by other countries. I think the international perspective of the manuscript is seriously insufficient, not only reflected in the introduction, so strengthening the comparison with wetlands in other countries can enhance the research significance of this article.

-We dealt with a case of wetland protected areas with a high proportion of private lands, called inholdings. In this situation, PES would be a feasible alternative. In many countries, there should be similar problems with inholdings in protected areas. In the conclusion section, we indicated this point along with several references that presented PES as a means to resolve the conflicts caused by private lands.

-As suggested, we inserted PES and similar programs employed in other countries in the literature review section. Also, we added a comparison of this study’s findings to show the effectiveness in other countries.

Finally, many details of the manuscript need to be supplemented, such as the chart design is not standardized (three-line table), the questionnaire design needs to be displayed, and the validity of the questionnaire, etc. Moreover, the language expression of the article needs to be improved, and further polishing is recommended.

-We made changes as suggested. Thank you.

-In a separate file (Supplementary Materials), we provided the instructions and questions used in the questionnaire.

-In terms of the validity of the questionnaire, we designed the CE questions based on the consultation with experts and a series of pretests. We also added a comparison of WTP and WTA in this study with similar programs to confirm that they are in a reasonable range. Thank you for the constructive comments.

Reviewer 3 Report

PES is a viable solution for protected areas to deal with conflicts or social problems from acquisition of private lands. The study estimated the beneficiaries’ WTP and providers’ WTA by employing the choice experiment method in wetland protected area of Korea and proposed reasonable contract standards for PES . Compared with the single perspective of previous studies, this study design considering the preferences of both and beneficiaries and providers is innovative. However, there are still some problems in the structure of the manuscript and the survey sampling method. Here are the suggestions.

1. The study design (Line 233-287) and data collection (Line 289-299) may be adjusted to the third part 'Materials and methods'. In addition, it is better to draw a diagram of study design and process to clearly show the difference from other studies.

2. For the data collection part, the survey sampling method is not clearly described. Only 52 providers were interviewed, and 33 providers' questionaries were analysed in the manuscript. Can 33 providers' response represent the stakeholders? The discussion is insufficient.

Author Response

  1. The study design (Line 233-287) and data collection (Line 289-299) may be adjusted to the third part 'Materials and methods'. In addition, it is better to draw a diagram of study design and process to clearly show the difference from other studies.

As suggested, we put ‘Study designs’ and ‘Data collection’ under ‘Methods’. Also, we added the process of the study (Figure 2) in the Study design (subsection 3.3).

  1. For the data collection part, the survey sampling method is not clearly described. Only 52 providers were interviewed, and 33 providers' questionaries were analysed in the manuscript. Can 33 providers' response represent the stakeholders? The discussion is insufficient.

-In this study, the providers were those who lived in the study areas and who participated in a direct payment program for eco-friendly farming. As a result, less than 100 farmers were eligible for the provider group, and we had contacted about 60% of them. Also, the average age of the providers was 67 and most of them had less than elementary school graduation. Although we made an effort to reach them, it was not possible to recruit all of them given the factors mentioned above. We think 33 providers are representative of the group, but we are also aware that this could be a limitation of the study. Therefore, we mentioned this as one of the limitations in the conclusion section.

Round 2

Reviewer 2 Report

Thanks to the author for his replies and revisions, which greatly improved the quality of the manuscript.

Reviewer 3 Report

The revised manuscript is more clear on the research design and sample selection.